# Multi-Class Freeway Congestion and Emission Based on Robust Dynamic Multi-Objective Optimization

**Juan Chen** [1,2,*] **, Qinxuan Feng** [1] **and Qi Guo** [1]

1   SHU-UTS SILC Business School, Shanghai University, Shanghai 201899, China;
    deepblue00@shu.edu.cn (Q.F.); iguoqi@i.shu.edu.cn (Q.G.)
2   Smart City Research Institute, Shanghai University, Shanghai 201899, China
*   Correspondence: chenjuan82@shu.edu.cn; Tel.: +86-10-6998-0028

**Abstract:** In order to solve the problem of traffic congestion and emission optimization of urban multi-class expressways, a robust dynamic nondominated sorting multi-objective genetic algorithm DFCM-RDNSGA-III based on density fuzzy c-means clustering method is proposed in this paper. Considering the three performance indicators of travel time, ramp queue and traffic emissions, the ramp metering and variable speed limit control schemes of an expressway are optimized to improve the main road and ramp traffic congestion, therefore achieving energy conservation and emission reduction. In the VISSIM simulation environment, a multi-on-ramp and multi-off-ramp road network is built to verify the performance of the algorithm. The results show that, compared with the existing algorithm NSGA-III, the DFCM-RDNSGA-III algorithm proposed in this paper can provide better ramp metering and variable speed limit control schemes in the process of road network peak formation and dissipation. In addition, the traffic congestion of expressways can be improved and energy conservation as well as emission reduction can also be realized.

**Keywords:** robust dynamic; nondominated sorting multi-objective genetic algorithm; multi-class; traffic congestion and emissions

## 1. Introduction

The urban expressway system is the main framework system of the urban road system, and its operation service level directly affects the efficiency of urban road traffic. Taking Beijing, Shanghai and other big cities in China as an example, the problem of congestion on expressways is becoming increasingly serious [1]. At the same time, the on and off ramps of urban expressways are also closely related to the traffic efficiency of the main road. In the entrance and exit area of the ramps, the impact of the incoming area of the ramp on the mainline traffic can be reduced by active management and control of the incoming vehicles. Therefore, the traffic efficiency of the expressway can be improved, and the emission as well as fuel consumption can be reduced. Relevant studies show that these vehicle exhaust pollutants are one of the main factors causing urban haze [2].

Since the 1870s, abundant research results have been achieved by scholars in the field of expressway traffic control. Taking the traffic control strategy as an example, many countries already have quite scientific variable speed limit control technology. However, research in China is still lagging behind and is unable to take advantages of variable speed limit control technology. From the beginning of the 1990s, when variable speed limit control technology was put into practice, to now, despite the ramp control technology currently implemented in Shanghai, the ideal control effect still cannot be obtained. Therefore, this paper will focus on better traffic control by proposing a more practical algorithm.

Active management and control of multi-class expressway systems are studied in this paper. Through adopting effective control strategies, congestion and traffic efficiency can be improved, and vehicle exhaust emissions and fuel consumption can be reduced. In addition, considering the dynamic variation of the traffic process, it is particularly

important to optimize control strategies while balancing both robustness and dynamic requirements and to implement a control strategy with strong stability and adaptability to the dynamic variation of traffic flow. Based on the above considerations, a robust dynamic nondominated sorting multi-objective genetic algorithm, DFCM-RDNSGA-III, based on densities and the fuzzy C-means clustering method is proposed in this paper. A robust control strategy for the multi-on-ramp expressway network can then be provided; this strategy can help solve the congestion situation in the peak h of the road network and make a positive contribution to environmental protection.

Specifically, DFCM-RDNSGA-III is designed to optimize the variable speed limit strategy and ramp control scheme of urban expressway networks. Considering the three performance indicators of travel time, ramp queue and traffic emissions, it combines the improved multi-class freeway macroscopic traffic flow model Multi class METANET with the multi-class emission and fuel consumption model Multi-class VT-macro. The variable speed limit and ramp metering strategy of the expressway system with multi-on-ramps and multi-off-ramps are optimized, and finally, the effect of the algorithm is verified through building road network under the VISSIM simulation environment.

The contributions of this paper are as follows:

1. A new robust dynamic multi-objective optimization algorithm is designed to simultaneously optimize the travel time, ramp queue and traffic emissions of expressway networks. In the proposed robust dynamic multi-objective optimization algorithm, the density fuzzy c-means clustering algorithm DFCM is adopted to explore the irregular Pareto fronts in the multi-objective optimization algorithm. This can lead to more reasonable reference point setting, therefore improving the diversity of Pareto solutions and accelerating the convergence speed of the optimization algorithm;

2. Based on the improved multi-class freeway macroscopic traffic flow model, multi-class METANET and the multi-class emission and fuel consumption model ulti-class VT-macro, the algorithm can optimize the variable speed limit and ramp metering rate of the expressway system with multiple on-ramps and off-ramps at the same time. Moreover, the ramp inflow rate and main road variable speed limit control scheme can be obtained with strong robustness and adaptability to the dynamic variation of traffic flow. Therefore, the main road and ramp traffic congestion can be improved at the same time and energy conservation as well as emission reduction can be achieved;

3. By building the actual road network in the VISSIM simulation environment, the algorithm proposed in this paper is compared with the existing algorithms. It is verified that the proposed algorithm, DFCM-RDNSGA-III, can provide better traffic control strategies, and the freeway traffic congestion and environmental pollution can be improved effectively.

## 2. Literature Review

Considering the problems solved in this paper, a literature review will be conducted below from four aspects: freeway models, freeway traffic control strategies, robust dynamic multi-objective optimization algorithms and Pareto front exploration.

Traffic flow theory is fundamentally applied in fields such as intelligent traffic system, traffic management and control, traffic engineering design, and so on [3]. In general, traffic flow models can be divided into microscopic and macroscopic models. Macroscopic models often contain fewer parameters than microscopic models, which will be used in this paper. Partial differential equations are used to describe the macroscopic traffic flow models. These macroscopic traffic flow models can be classified into first-order, second-order or higher-order models. Some models, such as the Lighthill–Whitham–Richards (LWR) [4] and cell transmission model (CTM) [5], are representative of first-order models, while the METANET model is a common second-order traffic flow model [6]. Since the speed variations of vehicles and the reaction time of drivers are not considered, capacity decreases and oscillations due to freeway congestion cannot be reflected in first-order models. However, such deficiencies can be remedied in second-order traffic flow models, since second-order models can reproduce highly realistic traffic phenomena. In second-

order models, the real traffic networks can be reproduced by the METANET model more accurately. For instance, the paper [7] extended a multi-class macroscopic traffic flow model, METANET, and proved that the model can lead to better fitting results because the heterogeneity of the traffic flow network structure was taken into account. Based on the above, this paper integrates the multi-class model proposed in paper [7] and a dynamic model proposed in paper [8]; therefore, a more comprehensive multi-class macroscopic traffic flow model is proposed.

Variable speed limits (VSL) and ramp metering (RM) are the most common strategies used in traffic control [9]. Based on fixed speed limits, VSL is a mainline control strategy where both frequent and incidental traffic congestion are considered. By using this strategy, dynamic speed limit suggestions can be provided to drivers in advance; therefore, the mainline flow upstream of the bottleneck can be well-managed to maximize the traffic throughput and prevent traffic congestion effectively [10]. In paper [7], a VSL-based cell transmission model was proposed to control the main roadway of a freeway, and the travel time was found to be reduced effectively. RM considers the traffic conditions and capacity on the downstream of the mainline. By using this strategy, the vehicle proportion of the ramp inflow can be determined without causing mainline traffic breakdown, therefore improving the network operation efficiency [11]. However, freeways still cannot be well-managed and controlled based on a single traffic control strategy due to the sophistication of expressway systems. Therefore, integrated control strategy is one of the present research directions [12]. The main idea is to regulate the expressway through coordinating more than one traffic control method. In paper [13], an integrated strategy was developed to coordinate multiple on-ramps and maximize the motorway bottleneck throughput. The limitations of the model in paper [13] include that only the throughput was taken into account, while the differences in vehicle classes were neglected. In conclusion, there are few studies on the integrated control strategy of VSL and RM on the aspect of multi-class expressways. Therefore, the research in this paper is meaningful.

In the process of optimizing practical problems, normally the original problem will be described as dynamic optimization problems (DOPs) when the environment changes dynamically [14]. In general, due to the real-time traffic flow variations on the entrance of the mainline and ramps, real-time oscillations often occur on the aspect of the traffic network states, such as density, flow, and speed, while dealing with the optimization problems of multi-class freeway congestion and emissions simultaneously. Therefore, freeway network control schemes need to be changed accordingly. The real-time variations of the demands, the network states and the control schemes were denoted in the form of parameters or performance indicators in DOPs. Moreover, the paper [15] described that historical environmental conditions and corresponding optimal solutions needed to be recorded and stored in the optimization process of DOPs. Therefore, obvious improvements in the evolution efficiency and search performance of the algorithm can be obtained through reusing these reserved solutions when detecting a similar dynamic environment. At present, robust dynamic optimization methods have been adopted by many researchers to solve DOPs, mainly including dynamic single-objective methods [16] and static multi-objective robust optimization methods [17]. However, research on robust dynamic multi-objective optimization methods is still limited. Hence, it is necessary to extend the research on robust dynamic multi-objective optimization algorithms extensively. While dealing with DOPs, the major challenge is to determine the distribution of Pareto fronts because of the irregular Pareto optimal front distribution nature. This is due to the fact that the robust dynamic solution sets, which constitute the Pareto optimal front, often change with the environment during the robust dynamic optimization process. Therefore, the research on the exploration of Pareto fronts will be reviewed below.

In paper [18], it is pointed out that clustering methods can help multi-objective optimization algorithms describe the Pareto front, detect the distribution of the solutions over the objective space, track the Pareto front continuously and guide search directions by the use of cluster centers, thereby leading to an increase in population diversity and

convergence speed. The k-means method, proposed in paper [19], is one of the most widely used clustering algorithms which belongs to hard clustering. The principle of hard clustering is to divide data points into specific clusters, and the corresponding relationship between data points and clusters is unique. In paper [20], the k-means clustering algorithm was used to determine the irregular Pareto front shapes in a multi-objective optimization algorithm. By dividing the population into several cluster sets, the overall structure of the Pareto front can be discovered, which makes the population evolve in different cluster centers, thereby improving the diversity and convergence speed of the NSGA-II algorithm. However, the number of initial cluster centers in the k-means method needs to be determined by empirical methods. Moreover, random generation of initial cluster centers may lead to unstable clustering results. It is proposed in paper [21] that fuzzy clustering is one of the most widely used soft clustering algorithms which can accurately describe the mediation of the data samples. Among the fuzzy clustering methods, the fuzzy c-means clustering method, which combines traditional clustering algorithms with fuzzy theory, can provide more flexible clustering results [22]. Considering the large dimension of the objective spaces in the multi-objective optimization problems, in most cases, the solutions contained in the Pareto front are difficult to classify into a specific cluster center. Therefore, in this case, it is not reasonable to use hard clustering methods to divide the solutions into a specific cluster. Accordingly, it might be a better choice to use soft clustering methods instead to divide the samples into each cluster center according to a certain proportion. However, there are still two issues which need to be remedied. First, the fuzzy c-means clustering algorithm needs to initialize the cluster center, which will affect the clustering results. If the generated initial cluster centers are relatively poor, the algorithm may be trapped in local optima rather than obtain global optima. Second, when solving high-dimensional optimization problems, some marginal solutions may not be effectively covered by the randomly initialized cluster centers, which may reduce the diversity of solutions. In conclusion, the research on using clustering methods to determine the Pareto front is still not suitable. More in-depth research needs to be carried out due to the fact that the accuracy of the final Pareto front will be affected by the different selections of clustering algorithms and cluster centers.

## 3. Robust Dynamic Multi-Objective Optimization Problem in Freeway Traffic Congestion and Emission

Normally the optimization problem in the traffic field has the characteristics of high dimension, dynamic variation, and certain stability of the implemented optimization scheme. In this paper, the traffic congestion and emission optimization problems of multi-class expressways are described as robust dynamic multi-objective optimization problems. The traffic efficiency and environmental effects of expressways are measured based on the three indexes of travel times, traffic emissions and ramp queues. All the abbreviations can be found in Table A1 and relevant notations are concluded in Table A2.

### 3.1. Robust Dynamic Performance Indicators

When solving robust dynamic multi-objective optimization problems, the objective function can be designed to ensure the robustness of the solutions in a continuously changing environment. In this paper, the performance indicators are designed as follows in the robust dynamic multi-objective optimization problems of multi-class freeway congestion and emission.

3.1.1. Robust Dynamic Travel Times

The length of temporal windows is defined as $T$, and the number of the temporal windows is denoted as $A$. The robustness of travel times in continuous variable temporal windows with each length $T$ and total number $A$ can be demonstrated by this performance indicator. It can be denoted as the average travel time spent in the predesigned temporal

windows, where two parts are included: the travel time spent in the mainstream and the waiting time spent at the ramps. Therefore, the objective function can be defined as:

$$\min J_1(k) = 1/A \sum_{t=0}^{A-1} \sum_{k} T \cdot \left[ \sum_{i} \sum_{c} \rho_{i,c}(k) \cdot L_i \cdot \lambda_i + \sum_{o} \sum_{i} \sum_{c} l_{o,i,c}(k) \right], \tag{1}$$

where $c$ denotes the vehicle classes, $i$ denotes the indexes for freeway sections and the freeway on-ramps are indicated by $o$. The time horizon is divided into $K$ time steps, and $k = 0, 1, 2, \ldots, K$ indicates the temporal step counter, which relates to the time $kT$ with sample time interval $T$ (h). In addition, $l_{o,i,c}(k)$ is the queue length of class $c$ in section $i$ of on-ramp o in time interval $(kT, (k + 1)T))$. $\rho_{i,c}(k)$ is the traffic density of class $c$ in section $i$ in time interval $(kT, (k + 1)T))$. Additionally, $L_i$ and $\lambda_i$ represent the length of section $i$ and the number of lanes in section $i$, respectively.

### 3.1.2. Robust Dynamic Traffic Emissions

To calculate robust dynamic traffic emissions, the length of temporal windows is denoted as $T$, and the number of them considered by decision maker is described as $A$ analogously. This indicator is denoted as the average emissions generated by the total vehicles in $A$ continuous variable temporal windows with the length $T$. This objective function can be calculated as:

$$\min J_{2y,c}(k) = 1/A \sum_{t=0}^{A-1} \left\{ \sum_{k} \sum_{i} \sum_{c} (J_{y,i,c}^t(k) + J_{y,i,c}^s(k)) + \sum_{k} \sum_{o} \sum_{c} \sum_{o \in Ramp} J_{y,on,o,c}(k) \right\} \tag{2}$$

where $A$ denotes the total number of temporal windows and $k$ is the time step counter.

The freeway stretch is divided into $i$ sections, and the vehicle classes are indexed by $c$. In addition, the index $y \in \{CO, HC, NO_x, FC\}$ denotes the set of all the emission categories. $J_{y,i,c}^t(k)$ indicates the emissions generated by class $c$ travelling in section $i$ in time interval $(kT, (k + 1)T)$, in which the superscript $t$ represents the normal travelling state. $J_{y,i,c}^s(k)$ denotes the emissions generated by class $c$ waiting in section $i$ in time interval $(kT, (k + 1)T)$, in which the superscript $s$ denotes the stopping state. $J_{y,on,o,c}(k)$ represents the emissions generated by class $c$ at on-ramp o in time interval $(kT, (k + 1)T)$, in which the on-ramp is denoted as on and indexed by the notation $o$.

### 3.1.3. Robust Dynamic Ramp Queues

Similarly, the temporal window length is denoted as $T$, and the number of these temporal windows is described as $A$. Robust dynamic ramp queues can be denoted as the average number of vehicles waiting at the on-ramps in the continuous variable temporal windows. This performance indicator can be expressed as:

$$\min J_{3i,c}(k) = 1/A \sum_{t=0}^{A-1} \sum_{k} \sum_{c} \sum_{i} (l_{i,c}(k) + T[d_{i,c}(k) - r_{i,c}(k)]), \tag{3}$$

where $k$ denotes the time step counter, $i$ denotes the on-ramp $i$ of the freeway and the total number of temporal windows is described as $A$. The length of the temporal window or the sample time interval can be denoted as $T$ (h). Additionally, $l_{i,c}(k)$ denotes the queue length of vehicle class $c$ at on-ramp $i$ in time interval $(kT, (k + 1)T)$. $d_{i,c}(k)$ indicates the allowed traffic volume of class $c$ entering mainstream at on-ramp $i$ in time interval $(kT, (k + 1)T)$, whereas $r_{i,c}(k)$ represents the actual traffic volume of class $c$ entering the mainstream at on-ramp $i$ in time interval $(kT, (k + 1)T)$.

### 3.2. Constraint

The detection of environmental variations, traffic safety requirements, and the range of variable values to meet actual demands are the three main constraints of this robust

dynamic optimization problem of multi-class, multi-on-ramp, and multi-off-ramp freeway congestion and emission.

### 3.2.1. Environmental Variation Detection

It is judged whether the entering traffic volume at the on-ramps exceeds the variable range compared with the former time instant. This can also be expressed as the variations of $r_{i,c}(k)$, the allowed on-ramp traffic volume of vehicle class $c$ entering the mainstream, which can be computed as:

$$r_{i,c}(k+1) - r_{i,c}(k) \geq \sigma \tag{4}$$

$$r_{i,c}(k) = \mu_{i,c}(k)\bar{r}_{i,c}(k), \tag{5}$$

where $r_{i,c}(k)$ denotes the allowed on-ramp traffic volume of class $c$ entering section $i$ at time instant $kT$, which can be obtained as the ramp control flow of the last time in the optimization algorithm. $\bar{r}_{i,c}(k)$ indicates the demanded on-ramp traffic volume of class $c$ entering section $i$ at time instant $kT$, and $\mu_{i,c}(k)$ is the metering rate, which can also be described as the portion of the flow allowed to enter the mainstream under ramp metering. In addition, three on-ramps are set in this paper; therefore $i = 1, 2, 3$, and $\sigma$ denotes the threshold of the detection, which is set equal to 0.1 in this paper.

It is also judged whether the entering traffic volume in the mainstream exceeds the variable range compared with the last time, which can be expressed as:

$$q_{i,c}(k+1) - q_{i,c}(k) \geq \sigma, \tag{6}$$

where $q_{i,c}(k)$ denotes the traffic volume of class $c$ entering section $i$ in time interval $(kT, (k+1)T)$. Analogously, $\sigma$ denotes the threshold of the detection. Variations are regarded to occur in the environment if the difference between two adjacent times exceeds the setting threshold. Additionally, the threshold $\sigma$ is suitable to be set equal to 0.1 by multiple trials.

### 3.2.2. Traffic Safety Requirements

It is required that the variable range of the two variable speed limits should be constrained within 20 km/h, so that traffic incidents can be avoided though preventing the two continuous variations from being too large. It can be expressed as:

$$VSL_{c,i}(k+1) - VSL_{c,i}(k) \leq 20, \tag{7}$$

where $VSL$ can be described as the value of the variable speed limit and $VSL_{c,i}(k)$ denotes the variable speed limit of vehicle class $c$ in section $i$ at time $kT$.

It is also required that the difference between two adjacent sections should be constrained within 20 km/h, so that traffic breakdowns, such as rear-end collisions, can be avoided through preventing the deviation from being overlarge. It can be defined analogously as:

$$VSL_{c,i+1}(k) - VSL_{c,i}(k) \leq 20, \tag{8}$$

### 3.2.3. Range of Variables

The range of variables is required to meet the actual demands; therefore in this paper, the $VSL$ of cars is set between 60 and 90 km/h and the $VSL$ of trucks is set between 50 and 80 km/h, which can be expressed respectively as: $VSL_1 \in (60, 90)$ (km/h) and $VSL_2 \in (50, 80)$ (km/h). Additionally, both the ramp metering rates are set in the range $(0, 1)$.

## 4. The Robust Dynamic Nondominated Sorting Multi-Objective Genetic Algorithm Based on Density Fuzzy C-Means Method DFCM-RDNSGA-III

The nondominated sorting multi-objective genetic algorithm NSGA-III proposed in paper [23] is mainly used to solve traditional static multi-objective optimization problems. This paper needs to solve the robust dynamic multi-objective optimization control problems

of the freeway system. Therefore, the original NSGA-III algorithm needs to be improved to meet the requirements of providing robust solutions in a dynamic environment. At the same time, due to the difficulty in determining the Pareto fronts in the optimization process, a clustering algorithm is used in this paper to obtain the specific distribution of Pareto fronts. The clustering centers are used to describe the Pareto fronts and detect the distribution of the solutions in the target space. Through continuously tracking the Pareto fronts, the search is guided forward to the optimal Pareto fronts, and the population diversity and convergence speed can also be increased. Therefore, this paper improves the nondominated sorting multi-objective genetic algorithm NSGA-III in paper [23], and a robust dynamic nondominated sorting multi-objective genetic algorithm, RDNSGA-III, is proposed based on the concept of robust optimization and dynamic optimization.

Based on the original NSGA-III algorithm, the environment detection operator is introduced to store the environment information and the set of optimal solutions corresponding to the current environment. Additionally, the robust dynamic objective function is set to meet the needs of robust dynamic optimization. Secondly, in order to solve the problem of the difficulty of determining the Pareto fronts in the optimization process, a density fuzzy c-means clustering algorithm DFCM is proposed to determine the distribution of Pareto fronts. This algorithm can detect the distribution of solutions in the target space, track the Pareto front continuously, and guide the search direction of the algorithm. Finally, combining the robust dynamic optimization algorithm given in this section with density fuzzy c-means clustering algorithm DFCM, a robust dynamic nondominated sorting multi-objective genetic algorithm based on the density fuzzy c-means clustering algorithm DFCM-RDNSGA-III is proposed. The details of the proposed algorithm are shown in the following.

### 4.1. Robust Dynamic Nondominated Sorting Multi-Objective Genetic Algorithm RDNSGA-III

In the robust dynamic nondominated sorting multi-objective genetic algorithm RDNSGA-III proposed in this section, there are two main improvements:

1. In terms of dynamic characteristics, an environment detection operator is introduced based on the NSGA-III algorithm to test whether the external environment has changed. If environmental variations are detected, optimization will be restarted to obtain the optimal solutions which satisfy the current environment. Moreover, in the process of the algorithm operating, the environment information and the optimal solution sets corresponding to the current environment will be stored. It is ensured that repeated optimization under a similar environment can be avoided during the dynamic optimizing process;

2. In terms of robustness, the robust dynamic objective function is introduced to measure the robustness of the solutions in the nondominated sorting, i.e., the robust dynamic objective function is used to sort the solutions. By minimizing the average value of the objective function in multiple continuous temporal windows, the solution with stronger robustness is not only applicable to the current dynamic environment, but also applicable to multiple continuous dynamic environments. The robust dynamic objective function setting has been described in detail in Formula (1) to Formula (3) above.

The framework of the RDNSGA-III algorithm is described as follows:

Step 1: Initialize the environment detection parameter, set the environmental detection counter $t = 1$, the maximum environmental detection times $t_{max}$, and $bestpop_{final} = \varnothing$ to store the current environment information and corresponding optimal solutions.

Step 2: If the environment detection counter $t = 1$, or if the environment detection counter $t > 1$ and the environment has changed, go to step 3. Otherwise, copy the current population $P_1$, and go to step 12.

Step 3: Initialize algorithm parameters, including the maximum iterations $Gen_{max}$, the number of the population $pop$, the current generation $gen = 0$, initialize the population $P_0 = \{x(1), \ldots, x(pop)\}$, nondominated solution set $S_t = \varnothing$, archive set $D_t = \varnothing$, and reference point set $Z^s = \varnothing$.

Step 4: Recombine, crossover and mutate the population $P_t$ to generate offspring population $Q_t$, $Q_t = Recombination(P_t) + Mutation(P_t)$.

Step 5: Combine archive set $D_t$ and offspring population $Q_t$ to generate the combined population $R_t$, $R_t = D_t \cup Q_t$.

Step 6: Perform the nondominated sorting operation on the combined population $R_t$ and generate the nondominated solution set $R_t = \{F_1, F_2, F_3, \ldots, F_l, \ldots\}$. Perform nondominated sorting on the nondominated solution set $R_t$ to obtain the set $U_t$, $U_t =$ Non-dominated-sort$(R_t)$, where $F_1$, $F_2$, $F_3$, $\cdots$, $F_l$, $\cdots$ denotes the nondominated solution sets with nondomination level $1, 2 \ldots l, \ldots$, respectively, $F_1 \succ F_2 \succ F_3 \succ \cdots \succ F_l \succ \cdots$.

Step 7: Generate nondominated solution set $S_t$.

Step 8: Generate the next-generation population $P_{t+1}$. If the number of solutions in $S_t$ is exactly equal to $N$, i.e., $|S_t| = N$, the next generation of parent population $P_{t+1}$ is generated directly, $P_{t+1} = S_t$, and $t = t + 1$, return to step 2; otherwise, structure $P_{t+1}$ with $F_1, F_2, \ldots, F_{l-1}$, i.e., $P_{t+1} = \cup_{j=1}^{l-1} F_j$. The remaining $K$ solutions need to be selected from the layer $F_l$ according to the niche count, i.e., the number of solutions that the population $P_{t+1}$ needs to select from $F_l$ is $K = N - |P_{t+1}|$.

Step 9: Generate reference points; the reference point set $Z^r = Normalize(f^n, S_t, Z^s, Z^r)$.

Step 10: Associate the solutions in the solution set $S_t$ with the reference points.

Step 11: Calculate the niche count of each reference point $j$ in the reference point set $Z^r$, select elements according to the niche count to construct the population $P_{t+1}$, and select $K$ solutions from $F_l$ to join the population $P_{t+1}$.

Step 12: Reserve the existing population, randomly take out half of the solutions and store them in the archive set $D_{t+1}$. Let $gen = gen + 1$; if $gen < Gen_{max}$, go to step 4. if $gen = Gen_{max}$, output the final solution set $P_{final}$, and go to step 13.

Step 13: Save the current environment information and the final optimal solution $P_{final}$ to the optimal solution set $bestpop$, and add it to $bestpop_{final}(t)$. Let $t = t + 1$; if $t < t_{max}$, go to step 2. If $t = t_{max}$, output the final optimal solution set.

### 4.2. The Robust Dynamic Nondominated Sorting Multi-Objective Genetic Algorithm Based on Density Fuzzy C-Means Method DFCM-RDNSGA-III

In the paper [24], it is pointed out that the main challenge of solving the Pareto front distribution problem is the unknown distribution of Pareto optimal fronts in multi-objective optimization algorithms. The paper [25] pointed out that clustering algorithms can help multi-objective optimization algorithms detect the distribution of solutions in the objective space and guide the search to the Pareto fronts. However, the randomness of initial cluster generation and the difficulty of determining the number of initial cluster centers occur when using fuzzy c-means clustering algorithm. Therefore, a density-based clustering method (DBSCAN) [26] is adopted in this paper to conduct density clustering of the initial cluster centers when initializing the cluster centers.

The reasons to adopt this method can be explained from three aspects. Firstly, the spatial characteristics of data points can be reflected, and the rationality of cluster center distribution can be ensured when using density clustering, therefore improving the quality of the initial cluster centers of the fuzzy c-means clustering algorithm. Secondly, the number of cluster centers does not need to be set in advance because cluster centers can be generated automatically according to the specific density distribution of the population, which can solve the problem of setting the number of cluster centers by trial and error in fuzzy c-means clustering algorithm. In addition, through the fuzzy c-means clustering algorithm, the given initial cluster center set is continuously optimized, and the position of the cluster center is adjusted; therefore the shape information of more real Pareto fronts can be obtained.

Based on the RDNSGA-III algorithm in Section 4.1 and the DFCM algorithm proposed in paper [26], a robust dynamic nondominated sorting multi-objective genetic algorithm based on density fuzzy c-means method DFCM-RDNSGA-III is proposed in this section. The major improvement of this algorithm is that, in the following description of the

DFCM-RDNSGA-III algorithm, specifically in Steps 9 and 10, the difficulty of determining Pareto fronts during the algorithm optimization process is solved based on density fuzzy c-means method DFCM. Specific distribution of the Pareto fronts can be acquired through adopting this clustering method. By describing the Pareto fronts through cluster centers, the distribution of the solutions over the objective space is detected, and the Pareto fronts can be tracked continuously [27]. Therefore, the search direction can be guided towards the Pareto fronts, leading to an increase in population diversity and convergence speed. Additionally, the details of the improvement are presented in sub-algorithm 1 and sub-algorithm 2 in this section.

The flowchart of the proposed algorithm DFCM-NSGA-III is given in Figure 1. The framework of the proposed algorithm DFCM-NSGA-III will be described first, then sub-algorithm 1 and sub-algorithm 2 used in the proposed algorithm will be discussed later.

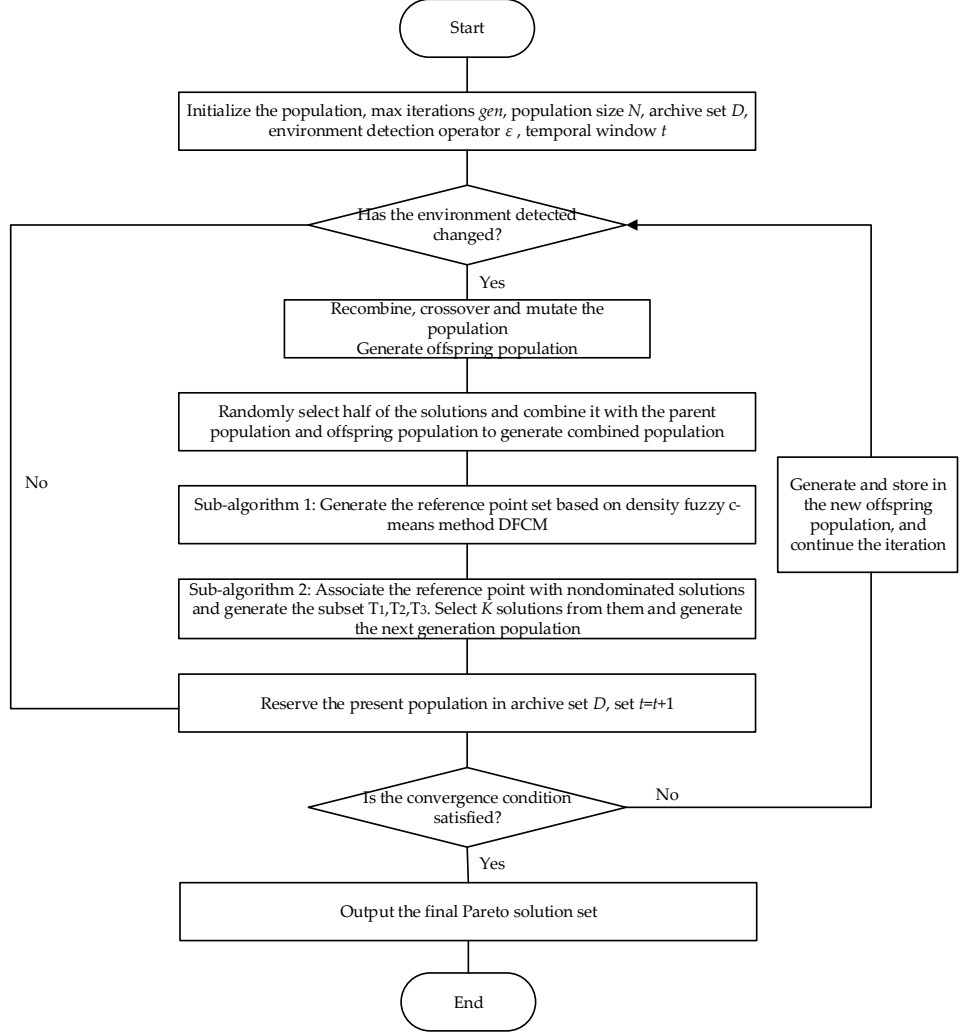

**Figure 1.** Flow chart of the algorithm DFCM-RDNSGA-III.

Specifically, the execution steps of the DFCM-RDNSGA-III algorithm are as follows:

Step 1: Initialize the environment detection parameter, set the environmental detection counter $t = 1$, the maximum environmental detection times $t_{max}$, and set $bestpop_{final} = \varnothing$ to store the current environment information and corresponding optimal solutions.

Step 2: Initialize the optimization process of the algorithm, or if $t > 1$ and the environment has changed, go to Step 3. Otherwise, copy the current population $P_1$, and go to Step 11.

Step 3: Initialize algorithm parameters, including the maximum iterations $Gen_{max}$, the number of population $pop$, the current generation $gen = 0$, the population $P_0 = \{x(1), \ldots, x(pop)\}$,

the nondominated solution set $S_t = \varnothing$, the archive set $D_t = \varnothing$, and the reference point set $Z^s = \varnothing$.

Step 4: Recombine, crossover and mutate the population $P_t$ to generate the offspring population $Q_t$, $Q_t = Recombination(P_t) + Mutation(P_t)$.

Step 5: Randomly select half of the solutions as the set $D_{past}$ from the current archive set $\{D_t\}$. Combine the parent population $P_t$ and offspring population $Q_t$ to generate the combined population $R_t$, $R_t = P_t \cup Q_t \cup D_{past}$.

Step 6: Perform the nondominated sorting operation on the combined population $R_t$, and generate the nondominated solution set $R_t = \{F_1, F_2, F_3, \ldots, F_l, \ldots\}$. Perform nondominated sorting on the nondominated solution set $R_t$ to obtain the set $U_t$, $U_t = $ Non-dominated-sort$(R_t)$, where $F_1, F_2, F_3, \cdots, F_l, \cdots$ denotes the nondominated solution sets with nondomination level $1, 2 \ldots l, \ldots$, respectively, $F_1 \succ F_2 \succ F_3 \succ \cdots \succ F_l \succ \cdots$.

Step 7: Generate nondominated solution set $S_t$.

Step 8: Generate the next generation population $P_{t+1}$. If the number of solutions in $S_t$ is exactly equal to $N$, i.e., $|S_t| = N$, the next generation of parent population $P_{t+1}$ is generated directly, $P_{t+1} = S_t$, and $t = t + 1$, go to Step 2; otherwise, construct $P_{t+1}$ with $F_1, F_2, \ldots, F_{l-1}$, i.e., $P_{t+1} = \cup_{j=1}^{l-1} F_j$. The remaining $K$ solutions need to be selected from the layer $F_l$ according to the niche count, i.e., the number of solutions that the population $P_{t+1}$ needs to be selected from $F_l$ is $K = N - |P_{t+1}|$.

Step 9: Generate reference points according to the density fuzzy c-means clustering algorithm DFCM and construct the reference point set $Z^r$. The specific process is described in sub-algorithm 1.

Step 10: Associate the solutions in the solution set $S_t$ with the reference points. Calculate the niche count of each reference point $j$ in the reference point set $Z^r$, select elements according to the niche count to construct the population $P_{t+1}$, and select $K$ solutions from $F_l$ as set $\{rest\}$ to join the population $P_{t+1}$. The specific process is shown in sub-algorithm 2.

Step 11: Reserve the existing population, randomly take out half of the solutions and store them in the archive set $D_{t+1}$. Let $gen = gen + 1$. If $gen < Gen_{max}$, return to step 4. If $gen = Gen_{max}$, output the final solution set $P_{final}$, and go to Step 12.

Step 12: Save the current environment information and the final optimal solution $P_{final}$ to the optimal solution set $bestpop$, and add it to $bestpop_{final}(t)$. Let $t = t + 1$; if $t < t_{max}$, go to Step 2. If $t = t_{max}$, output the final optimal solution set.

(1) **Sub-algorithm 1**: Calculate the cluster centers of $F_l$ and generate reference point set $Z^r$ according to the algorithm DFCM.

Step 1: Initialize the radius of neighborhood $\varepsilon$, and the minimum number of samples *MinPts*.

Step 2: Generate the initial cluster center set $\{Center_1\}$. Calculate the initial membership degree $\mu_{ij}$ of all solutions to the initial cluster center set $\{Center_1\}$ in the current situation, $\mu_{i,j} = \dfrac{1}{\sum_{k=1}^{H} \left( \frac{\|x_i - c_j\|}{\|x_i - c_k\|} \right)^{\frac{2}{m-1}}}$. $H$ denotes the number of cluster centers; $m(m > 1)$ denotes the fuzzy allocation matrix index used to control the degree of fuzzy overlap. $x_i$ refers to the $i$th solution and $c_j$ denotes the $j$th cluster center.

Step 3: Calculate the initial adaptiveness function $J$ of fuzzy c-means clustering algorithm, $J = \sum_{i=1}^{G} \sum_{j=1}^{H} \mu_{ij} \|x_i - c_j\|^2$, where $G$ denotes the number of all solutions in the $F_l$ and $H$ denotes the number of cluster centers.

Step 4: If the adaptiveness function $J$ does not reach the acceptable extent or if the maximum number of iterations is satisfied, go to Step 5; otherwise, end the procedure and output the clustering result.

Step 5: Update the cluster centers to form a new cluster center set $c_j = \dfrac{\sum_{i=1}^{D} \mu_{ij} x_i}{\sum_{i=1}^{D} \mu_{ij}}$.

Step 6: Update the membership degree $\mu_{ij}$ and adaptiveness function $J$ according to the new cluster centers and repeat Steps 3–5 until the output conditions are satisfied. Finally, output the final clustering results $\{Center\} = \{c_1, c_2, \ldots, c_H\}$, i.e., the reference point set $Z^r$.

Step 7: Generate the solution set $\{xc_i\}$ of the corresponding solutions included in the $i$th cluster center $c_i$ according to the cluster center set $\{Center\}$. Define the set of all the cluster centers and the solutions included as $\{xc\}$.

(2) **Sub-algorithm 2**: Solution selection method based on the density fuzzy c-means clustering algorithm DFCM.

Sub-algorithm 2 provides the solution selection method based on the density fuzzy c-means clustering algorithm DFCM. The purpose is to select $K$ solutions from $F_l$, $K = N - |P_{t+1}|$, and join them to the set $\{rest\}$ to construct the next generation population. The specific process is described as follows.

Step 1: Calculate the distance of all the solutions $\{x_1, x_2, \ldots, x_G\}$ in $F_l$ and each reference point in reference point set $Z^r = \{c_1, c_2, \ldots, c_H\}$. $Distance = \|x_i - c_j\|_2$, $x_i$ denotes the $i$th solution in $F_l$, while $c_j$ denotes the $j$th reference point in the reference point set $Z^r$.

Step 2: Initialize the number of all the solutions of $F_l$ distributed by $c_j$, $nc_j = 0$. $x_i$ being dominated by $c_j$ can be denoted as $x_i^j = \operatorname{argmin}(Distance(x_i, c_j))$. If $x_i$ is dominated by $c_j$, set $nc_j = nc_j + 1$.

Step 3: For $j = 1 : H$ reference points, if $nc_j > 0$, select the solution $x_{i_{min}}^j$ among all the solutions that has the closest distance to the reference point, and join it to set $T_1 = T_1 \cup \left\{ x_{i_{min}}^j \middle| \operatorname{argmin}\left(Distance\left(x_i^j, c_j\right)\right)\right\}$. If $nc_j \geq 3$, except the solutions in $T_1$, the remained solutions are put into the set $T_2 = T_2 \cup \left\{ x_i^j \middle| x_i^j \neq x_{i_{min}}^j \right\}$. Otherwise, add all the solutions except the solutions in $T_1$ and $T_2$ to the set $T_3 = T_3 \cup \left\{ x_i^j \middle| x_i^j \neq x_{i_{min}}^j \right\}$.

Step 4: If the number of the solutions in $T_1$ is equal to $K(K = N - |P_{t+1}|)$, i.e., $|T_1| = K$, then $\{rest\} = T_1$. If the number of solutions in $T_2$ is greater than or equal to $K$ subtracted by the number of solutions in $T_1$, i.e., $|T_2| > K - |T_1|$, then randomly select the remained solutions in $T_2$ and construct the solution set $\{T_2rest\}$, $\{rest\} = T_1 \cup \{T_2rest\}$. If the number of solutions in the set $T_3$ is greater than or equal to $K$ subtracted by the number of solutions in $T_1$ and $T_2$, i.e., $|T_3| > K - |T_1| - |T_2|$, then randomly select the remaining solutions in $T_3$ and construct the solution set $\{T_3rest\}$, $\{rest\} = T_1 \cup \{T_2rest\} \cup \{T_3rest\}$. Finally, output the set $\{rest\}$ of $K$ solutions selected from $F_l$.

Taking into account the computational complexity of the existing and proposed algorithms, the worst-case computational complexity of NSGA-III is $O\left(N^2 \log^{M-2} N\right)$, [23]. The algorithms RDNSGA-III and DFCM-RDNSGA-III are both proposed on the basis of NSGA-III. Therefore, the computational complexity of the two algorithms can be computed analogously by following the method in the paper [23]. The nondominated sorting of a population of size $2N$ having $M$-dimensional objective vectors requires $O\left(N^2 \log^{M-2} N\right)$(usually $M < N$). For DFCM-RDNSGA-III, calculating the cluster centers in step 9 requires $O(N^2)$ (sub-algorithm 1). The density determination of the reference points requires $O(N)$. In step 10 of DFCM-RDNSGA-III, the solution selection based on the density fuzzy c-means clustering method requires $O(MN^2)$ (sub-algorithm 2). Therefore, the computational complexity of DFCM-RDNSGA-III in the worst case is $O(N^2 \log^{M-2} N + MN^2)$, which is a little larger comparable to NSGA-III and RDNSGA-III.

## 5. Simulation

The simulation network and parameters are introduced in this section, then the traffic control strategy is described, and finally the experimental results are compared and analyzed.

### 5.1. Simulation Network

The multi-on-ramp and multi-off-ramp expressway network in paper [6] is adopted in this experiment. This freeway network has an 18 km main road, and there are three on-ramps $(O_1, O_2, O_3)$ and three off-ramps $(O_4, O_5, O_6)$. Two lanes are subsumed in the

main road, which are equally divided into 18 sections. Each section is 1 km. Additionally, the on-ramp and off-ramp are all single-lane ramps, and the length of both on-ramps and off-ramps is 1 km. The specific network diagram is as shown in Figure 2.

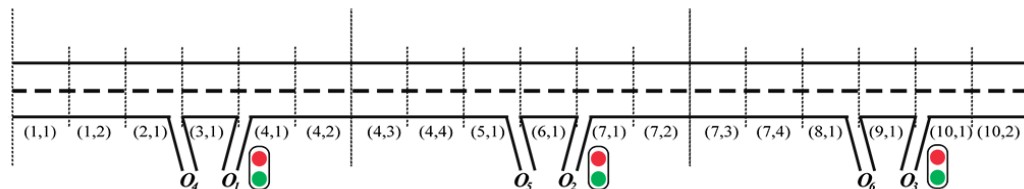

**Figure 2.** Structure of expressway.

*5.2. Network Overview and Parameter Setting*

5.2.1. Network Overview and Model Parameters

In order to simulate the peak hour curve of traffic inflow, the time span of the simulation time needs to be enough long to cover the whole process. Therefore, 2 h is the appropriate value. As for the sampling period, and control period, the experiment needs to collect enough data for calculation. On the one hand, it needs to quickly respond to changes in the current traffic flow. On the other hand, it cannot issue control commands very frequently, as this could result in traffic accidents. Based on the above considerations, the classical value is adopted, which can be seen in the paper [8]. To sum up, the simulation time is 2 h, the sampling period $T$ is 10 s and the control period is 5 min. The implemented control variables are adjusted according to the detection of environmental change and the robustness of the solution in the proposed DFCM-RDNSGA-III algorithm. The proportion of vehicles in the simulation road network is 80% cars and 20% trucks, assuming that the driver compliance rate for the control variables is 100%. The metering rate of the off-ramp is 10% of its adjacent mainline flow. The control parameter settings are shown in Table 1.

**Table 1.** Multi-ramp expressway network simulation parameters.

| Parameter Name | Parameter Value |
|---|---|
| Simulation time | 2 h |
| Sampling period ($T$) | 10 s |
| Control period ($KT$) | 5 min |

Table 2 is the actual value of the model simulation parameters. The multi-on-ramp and multi-off-ramp expressway network in paper [6] is adopted in this experiment, and all the actual values in Table 2 are set according to this expressway network. In Table 2, $\tau_1$ denotes the car lag time parameters, $\eta_1$ denotes the car correction coefficient, $v_1$ denotes the car expectation constant, $k_1$ denotes the car ramp influence coefficient, $\tau_2$ denotes the truck lag time parameters, $\eta_2$ denotes the truck correction coefficient, $v_2$ denotes the truck expectation constant, and $k_2$ denotes the truck ramp influence coefficient. In the freeway network, the real-time variations of the mainline car inflow and the truck inflow are shown in Figure 3. The real-time variations of the on-ramps ($O_1, O_2, O_3$) car and truck inflow are shown in Figure 4.

**Table 2.** Traffic flow model time-phased parameters.

| $\tau_1$ | $\eta_1$ | $v_1$ | $k_1$ | $\tau_2$ | $\eta_2$ | $v_2$ | $k_2$ |
|---|---|---|---|---|---|---|---|
| 0.315 | 60.00 | 25.00 | 0.45 | 0.3 | 65.00 | 20.00 | 0.40 |

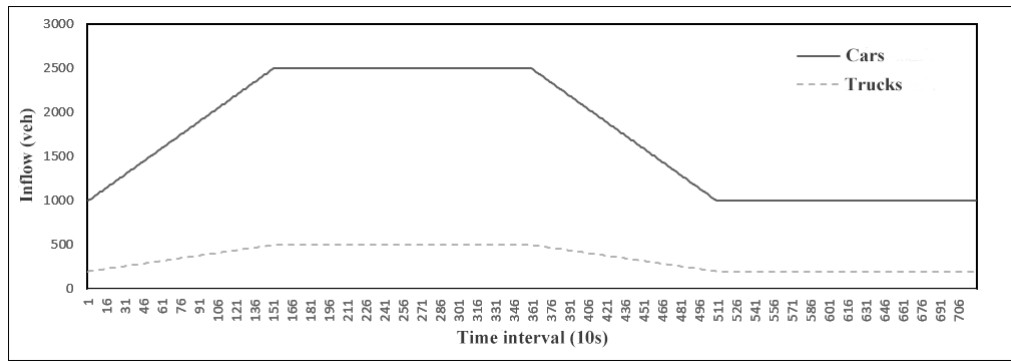

**Figure 3.** Inflow on the main road.

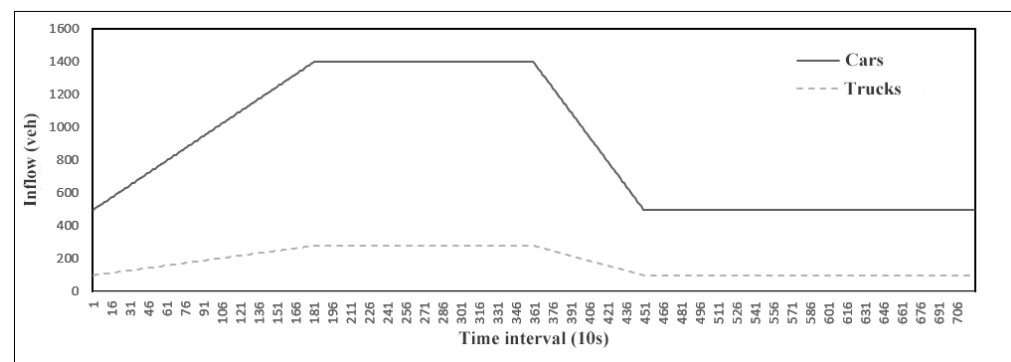

**Figure 4.** Inflow on the on-ramp.

5.2.2. Algorithm Parameter Setting

Two existing multi-objective optimization algorithms, NSGA-III and RDNSGA-III, are used in this paper for comparison of results with the proposed DFCM-NSGA-III algorithm. The parameter setting of the three algorithms are shown in Table 3, where the vacancy value indicates that the type of parameter is not involved in the corresponding algorithm. Among them, the NSGA-III algorithm is a static multi-objective optimization algorithm that optimizes the variable speed limit and the ramp control strategy without involving the environmental detection parameters. Through several tests, it is shown that a satisfactory solution with better robustness can be obtained if the number of continuously changed temporal windows is set to be 3. To ensure safety in actual traffic driving, the control period implemented by each variable speed limit is 5 min.

**Table 3.** Parameter setting.

| Parameters | NSGA-III | RDNSGA-III | DFCM-RDNSGA-III |
|---|---|---|---|
| Population size | 100 | 100 | 100 |
| Iteration numbers | 100 | 100 | 100 |
| Sampling period | 10 s | 10 s | 10 s |
| Environment detection threshold | - | 10% | 10% |
| Environment detection period | - | 5 min | 5 min |
| Temporal window numbers | - | 3 | 3 |
| DFCM iteration numbers | - | - | 100 |

*5.3. Control Strategy*

The variable speed limit of cars is set as (60, 90), the variable speed limit of trucks is set as (50, 80) and the ramp metering rate is set in the range (0, 1). A specific schematic diagram of control strategy implementation is shown in Figure 5, where VSL represents the section of variable speed limit control and ramp control indicates the on-ramp implementing the ramp control strategy.

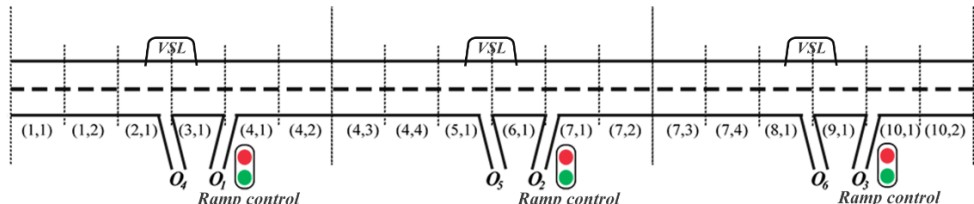

**Figure 5.** Schematic diagram of traffic control strategy implementation.

By adopting the robust dynamic multi-objective optimization algorithm, a set of optimal solutions will finally be obtained, therefore leading to the problem of choosing the optimal solution. Three performance indicators are considered in this paper: robust dynamic travel time, robust dynamic ramp queue and robust dynamic traffic emissions, which correspond to the patency degree of the main road, the patency degree of the ramps and the environmental influence. The optimal solution is selected by the weight method.

*5.4. DFCM Clustering Analysis*

Fuzzy c-means clustering is an unsupervised clustering method, so the following two indicators can be considered in judging its clustering effect:

1. The silhouette coefficient (SC) [28] is a way to evaluate the clustering effect, which can evaluate clustering results for different algorithms based on the same original data. The SC of clustering results is in the range $(-1, 1)$. The greater the value, the closer the same samples are and the further the different samples are, therefore leading to a better clustering effect.

2. The Davies–Bouldin Index (DBI) [29], also known as the classification and suitability indicator, was proposed by David Davies and Donald Bouldin to evaluate the effect of the clustering algorithms. This index measures the mean value of maximum similarity of each cluster. A smaller DBI value represents that the clustering results are close within the cluster interior, and the different clusters are farther separated; that is, the smaller the distance is within the class, and the greater the distance is between the classes.

The two indicators are calculated as follows:

$$SC = \frac{b(i) - a(i)}{\max\{a(i), b(i)\}}, \tag{9}$$

where $a(i)$ denotes the average distance from sample $i$ to other samples in the same cluster. The smaller it is, the more sample $i$ should be clustered into this cluster. It can be regarded as the dissimilarity of sample $i$ within the cluster. Similarly, $b(i)$ denotes the average distance from sample $i$ to all samples in other cluster centers $\{c_j\}$, which can be described as the dissimilarity between sample $i$ and other cluster centers $\{c_j\}$.

$$DBI = \frac{1}{N} \sum_{i=1}^{N} \max_{j \neq 1} \left( \frac{\overline{S_i} + \overline{S_j}}{\|w_i - w_j\|_2} \right), \tag{10}$$

where $N$ denotes the number of clusters. Variables $i$ and $j$ denote the cluster class $i$ and cluster class $j$. $\overline{S_i}$ and $\overline{S_j}$ indicate the average distance from the data of cluster class $i$ and cluster class $j$ to the cluster center, respectively, which represent the dispersion degree of samples in cluster class $i$ and cluster class $j$. $\|w_i - w_j\|_2$ denotes the distance between cluster class $i$ and cluster class $j$.

In this paper, DFCM-RDNSGA-III, a robust dynamic nondominated sorting genetic algorithm based on density fuzzy c-means clustering is proposed. The whole generated population in the iterative process of the algorithm is used as the test set. The clustering effect of the DFCM algorithm is compared with that of the basic fuzzy c-means clustering algorithm. The number of samples in the data set is 100, the number of iterations is 100, and the data dimension is 3. The initial number of cluster centers is set according to the

number of cluster centers generated by density clustering. The two algorithms are clustered 10 times, where EX is the mean of 10 times, DX is the variance of 10 times, DFCM denotes the density fuzzy c-means clustering algorithm, and FCM denotes the basic fuzzy c-means clustering algorithm. The comparison of indicators is shown in Table 4.

**Table 4.** DFCM actual clustering effect analysis.

| Number | SC | | DBI | |
|---|---|---|---|---|
| | DFCM | FCM | DFCM | FCM |
| 1 | 0.726883 | 0.561823 | 0.739189 | 0.624791 |
| 2 | 0.837247 | 0.767561 | 0.730389 | 0.853588 |
| 3 | 0.802877 | 0.712170 | 0.747989 | 0.791989 |
| 4 | 0.844320 | 0.625127 | 0.756789 | 0.695190 |
| 5 | 0.735415 | 0.640953 | 0.712790 | 0.712790 |
| 6 | 0.841398 | 0.648866 | 0.659990 | 0.721590 |
| 7 | 0.799813 | 0.696344 | 0.730389 | 0.774389 |
| 8 | 0.791686 | 0.815039 | 0.668790 | 0.906387 |
| 9 | 0.813086 | 0.783387 | 0.677590 | 0.871187 |
| 10 | 0.819470 | 0.720083 | 0.686390 | 0.800788 |
| EX | 0.797736 | 0.697135 | 0.711030 | 0.775269 |
| DX | 0.001502 | 0.005603 | 0.001112 | 0.006929 |

It can be seen from Table 4 that the density fuzzy c-means clustering algorithm DFCM has a better clustering effect than the basic fuzzy c-means clustering algorithm FCM in terms of SC and DBI among the 10 clustering results. From the 10 running results, the average score of the DFCM algorithm in SC increased by 14%, which shows that the distance between different samples in DFCM is further and the clustering effect is better. Compared with FCM, DFCM improves 8% on DBI, which shows that DFCM has a closer distance within clusters and further distance between clusters; that is, DFCM generates a better clustering effect. Secondly, DFCM is relatively stable. Comparing the variance of 10 results, it can be found that the variance of DFCM in the SC and DBI is increased by 73% and 84%, respectively, which indicates that DFCM can be more stable in the final clustering results, and it is difficult to have a large difference in clustering results. Therefore, the density fuzzy c-means clustering algorithm DFCM can deal with the overall structure distribution of Pareto fronts more stably and accurately.

*5.5. Results*

Each experiment was run for 5 times and the results of the three indicators are the average results. The results of three objective functions, including travel time, ramp queue, and traffic emission of the three algorithms, NSGA-III, RDNSGA-III and DFCM-RDNSGA-III, are compared in Table 5. Among these three algorithms, a static optimization framework is adopted in algorithm NSGA-III to statically optimize the variable speed limit and metering rate in the whole simulation, while a dynamic optimization framework is adopted to dynamically optimize these variables according to the environmental changes in the RDNSGA-III algorithm and the DFCM-RDNSGA-III algorithm.

It can be concluded from Table 5 that: (1) The travel time and traffic emission indicators performed the worst on the algorithm NSGA-III with a static optimization framework. Compared to NSGA-III, the two indicators decreased by 14.64% and 16.72%, respectively, in the algorithm RDNSGA-III, as well as 19.04% and 21.64%, respectively, in the algorithm DFCM-RDNSGA-III. (2) Although NSGA-III performs poorly on the travel time and traffic emission indicators, it performs the best on the aspect of the ramp queue indicator. Compared to NSGA-III, the ramp queue indicator increased by 20.51% and 30.36%, respectively, in the algorithm RDNSGA-III and DFCM-RDNSGA-III.

**Table 5.** Comparison of algorithm results.

| Algorithms | Travel Time (h) | Ramp Queue (veh) | Traffic Emission (kg) |
|---|---|---|---|
| NSGA-III | 2002.48 | 37.15 | 194.62 |
| RDNSGA-III | 1709.39 −14.64% | 44.77 +20.51% | 162.07 −16.72% |
| DFCM-RDNSGA-III | 1621.26 −19.04% | 48.43 +30.36% | 152.5 −21.64% |

The above comparison results show that, under the control scheme of NSGA-III, which has a static optimization framework, the ramp-entering vehicles are not limited when the main road capacity is relatively tight. Therefore, the pressure on the main road will be aggravated due to the excessive ramp-entering, which increases the total travel time and traffic emissions but decreases the ramp queue. For the algorithms under robust dynamic optimization frameworks, including RDNSGA-III and DFCM-RDNSGA-III, although the ramp queue of the DFCM-RDNSGA-III algorithm is improved by 9% compared with the former, the travel time and traffic emission are both improved by more than 5%. The reason is that the inflow of part of the ramp vehicles to the main road is sacrificed in the DFCM-RDNSGA-III algorithm, contributing to the improved traffic efficiency of the main road and the reduced environmental pollution caused by gas emissions.

The VSL and RM control schemes during the peak forming period and peak subsiding period of the RDNSGA-III and DFCM-RDNSGA-III algorithms are shown in Tables 6 and 7, while the niche detection threshold is set as $\varepsilon = 0.1$. The environment detection is carried out every 5 min. When it is detected that the environment has not changed, i.e., the threshold before and after the change is set within the predefined range, the optimization does not need to be performed again and the current control scheme continues to be used. This also means that the situation where the control scheme has not changed represents that the optimization need not to be restarted. Therefore, the robustness of the control scheme of the RDNSGA-III and DFCM-RDNSGA-III algorithms can be compared under the same environment detection threshold through the change of control schemes. A static framework is adopted in the basic algorithm NSGA-III to optimize the VSL and RM, so the change of the control scheme is not involved in it.

The variation in traffic control schemes of the RDNSGA-III algorithm and the DFCM-RDNSGA-III algorithm according to the change of the environment in the processes of peak hour and dissipation hour are shown in Tables 6 and 7. It can be seen that the average ramp metering rate of the RDNSGA-III algorithm at the three on-ramps is 68% within 0 min to 5 min of the peak formation process, and the ramp metering rate of the DFCM-RDNSGA-III algorithm is 48%. In the early stages of peak formation, the DFCM-RDNSGA-III algorithm begins to limit the ramp entry, so as to ensure the patency of the main road in the process of peak formation. After that, the ramp metering rate of the RDNSGA-III algorithm is still maintained at more than 60% within 5–15 min. At 15–20 min, it starts to limit the ramp entry, and the average ramp metering rate is set to 56%, while the ramp metering rate of the DFCM-RDNSGA-III algorithm is always set at about 50% during this period. Therefore, in the early peak formation process, the inflow of the DFCM-RDNSGA-III algorithm is maintained within the scope of environmental detection by limiting the ramp entry. While the ramp inflow in the RDNSGA-III algorithm is not restricted well in the early stage, the flow changes at the last and next time exceed the detection threshold. Therefore, within 20 min of peak formation, the control strategy is continuously adjusted before it starts to maintain stability. Similarly, RDNSGA-III has a certain conservative setting of VSL within 60 min to 75 min of peak dissipation. At the beginning of the peak dissipation process, the VSL is still controlled below 70 km/h. However, the DFCM-RDNSGA-III algorithm starts to maintain the VSL above 70 km/h after 60 min to 65 min of early peak dissipation process, therefore ensuring the rapid outflow of the main road in the process of peak dissipation. In general, the DFCM-RDNSGA-III algorithm has better control optimization stability than

the RDNSGA-III algorithm in the process of peak formation. The control strategy can be better dealt with in the process of peak formation, and solutions with better robustness can be generated under the same environmental sensitivity.

**Table 6.** Implementation of a variable speed limit control scheme.

| | Car VSL 1 | Car VSL 2 | Car VSL 3 | Truck VSL 1 | Truck VSL 2 | Truck VSL 3 |
|---|---|---|---|---|---|---|
| **RDNSGA-III** | | | | | | |
| **Time Interval** | | | **Peak Forming** | | | |
| 0–5 min | 76.03 | 66.93 | 61.67 | 65.25 | 68.05 | 68.17 |
| 5–10 min | 72.76 | 60.32 | 77.92 | 67.60 | 68.55 | 63.83 |
| 10–15 min | 70.73 | 77.00 | 62.84 | 66.70 | 63.15 | 66.34 |
| 15–20 min | 61.71 | 61.35 | 77.77 | 68.73 | 69.38 | 61.40 |
| 20–25 min | 60.03 | 60.06 | 61.75 | 63.41 | 65.41 | 69.26 |
| 25–30 min | 60.03 | 60.06 | 61.75 | 63.41 | 65.41 | 69.26 |
| **Time Interval** | | | **Peak Subsiding** | | | |
| 60–65 min | 64.14 | 71.52 | 77.39 | 61.03 | 65.47 | 0.79 |
| 65–70 min | 67.40 | 68.09 | 68.56 | 63.37 | 63.49 | 0.33 |
| 70–75 min | 67.40 | 68.09 | 68.56 | 63.37 | 63.49 | 0.33 |
| 75–80 min | 71.19 | 71.65 | 79.14 | 66.59 | 62.15 | 0.50 |
| 80–85 min | 71.19 | 71.65 | 79.14 | 66.59 | 62.15 | 0.50 |
| 85–90 min | 76.19 | 74.59 | 70.71 | 60.21 | 65.38 | 64.52 |
| **DFCM-RDNSGA-III** | | | | | | |
| **Time Interval** | | | **Peak Forming** | | | |
| 0–5 min | 79.28 | 62.31 | 61.03 | 69.01 | 65.41 | 64.32 |
| 5–10 min | 77.97 | 68.58 | 66.69 | 63.77 | 67.35 | 69.54 |
| 10–15 min | 77.97 | 68.58 | 66.69 | 63.77 | 67.35 | 69.54 |
| 15–20 min | 61.42 | 63.64 | 61.86 | 66.43 | 60.01 | 60.30 |
| 20–25 min | 61.42 | 63.64 | 61.86 | 66.43 | 60.01 | 60.30 |
| 25–30 min | 60.17 | 74.54 | 67.08 | 60.49 | 60.50 | 60.91 |
| **Time Interval** | | | **Peak Subsiding** | | | |
| 60–65 min | 64.36 | 66.51 | 76.59 | 64.53 | 62.92 | 63.56 |
| 65–70 min | 70.57 | 68.46 | 76.30 | 64.32 | 66.48 | 64.48 |
| 70–75 min | 70.57 | 68.46 | 76.30 | 64.32 | 66.48 | 64.48 |
| 75–80 min | 72.19 | 75.03 | 73.35 | 62.82 | 65.36 | 69.05 |
| 80–85 min | 74.09 | 77.49 | 60.55 | 67.26 | 65.63 | 65.86 |
| 85–90 min | 74.09 | 77.49 | 60.55 | 67.26 | 65.63 | 65.86 |

Figures 6 and 7 show the heat map of car speed and car flow in each road section of the three algorithms, in which the ordinate denotes the main road section number link 1 to link 18, and the abscissa denotes the time interval. In Figure 6, the speed heat map, it can be noticed through comparison that in the process of peak formation, the speed of each road section in DFCM-RDNSGA-III can be maintained at a relatively high level compared with the other two algorithms, which can better help control the ramp metering rate. Therefore, the traffic efficiency and vehicle speed of the main road can be improved. As can be seen from the flow heat map in Figure 7, the overall process of traffic flow is smoother from link1 to link18 in the DFCM-RDNSGA-IIII algorithm. The flow variation in each road section is within 500 veh, and the flow difference between each road section is small. However, the traffic flow of the NSGA-III algorithm and the RDNSGA-III algorithm have quite obvious and abrupt variations in the ramp merging section, especially at the ramp merging part of link 10. The traffic flow of NSGA-III algorithm suddenly changes from 2000 veh to 3000 veh, and the traffic flow of RDNSGA-III algorithm also increases by more than 500 veh. In the ramp merging section, the traffic flow surge indicates that in the

on-ramp area, there is too much traffic flow at the on-ramp entrance, which leads to traffic congestion due to a significant variation in the traffic flow of the front and rear sections.

**Table 7.** Implementation of ramp control strategies.

| | Car RM $O_1$ | Car RM $O_2$ | Car RM $O_3$ | Truck RM $O_1$ | Truck RM $O_2$ | Truck RM $O_3$ |
|---|---|---|---|---|---|---|
| | | | **RDNSGA-III** | | | |
| **Time Interval** | | | **Peak Forming** | | | |
| 0–5 min | 0.87 | 0.45 | 0.73 | 0.95 | 0.18 | 0.35 |
| 5–10 min | 0.72 | 0.49 | 0.77 | 0.55 | 0.54 | 0.31 |
| 10–15 min | 0.67 | 0.49 | 0.77 | 0.30 | 0.75 | 0.83 |
| 15–20 min | 0.58 | 0.55 | 0.56 | 0.51 | 0.75 | 0.30 |
| 20–25 min | 0.45 | 0.60 | 0.39 | 0.39 | 0.88 | 0.32 |
| 25–30 min | 0.45 | 0.60 | 0.39 | 0.39 | 0.88 | 0.32 |
| **Time Interval** | | | **Peak Subsiding** | | | |
| 60–65 min | 0.47 | 0.31 | 0.52 | 0.58 | 0.64 | 0.26 |
| 65–70 min | 0.52 | 0.20 | 0.76 | 0.33 | 0.68 | 0.22 |
| 70–75 min | 0.52 | 0.20 | 0.76 | 0.33 | 0.68 | 0.22 |
| 75–80 min | 0.63 | 0.34 | 0.74 | 0.41 | 0.44 | 0.73 |
| 80–85 min | 0.63 | 0.34 | 0.74 | 0.41 | 0.44 | 0.73 |
| 85–90 min | 0.66 | 0.75 | 0.58 | 0.63 | 0.86 | 0.38 |
| | | | **DFCM-RDNSGA-III** | | | |
| **Time Interval** | | | **Peak Forming** | | | |
| 0–5 min | 0.73 | 0.13 | 0.58 | 0.86 | 0.32 | 0.62 |
| 5–10 min | 0.55 | 0.35 | 0.69 | 0.28 | 0.20 | 0.37 |
| 10–15 min | 0.55 | 0.35 | 0.69 | 0.28 | 0.20 | 0.37 |
| 15–20 min | 0.56 | 0.43 | 0.77 | 0.27 | 0.21 | 0.84 |
| 20–25 min | 0.56 | 0.43 | 0.77 | 0.27 | 0.21 | 0.84 |
| 25–30 min | 0.39 | 0.38 | 0.91 | 0.74 | 0.23 | 0.88 |
| **Time Interval** | | | **Peak Subsiding** | | | |
| 60–65 min | 0.31 | 0.88 | 0.74 | 0.45 | 0.98 | 0.68 |
| 65–70 min | 0.54 | 0.98 | 0.77 | 0.90 | 0.50 | 0.83 |
| 70–75 min | 0.54 | 0.98 | 0.77 | 0.90 | 0.50 | 0.83 |
| 75–80 min | 0.66 | 0.31 | 0.57 | 0.49 | 0.32 | 0.49 |
| 80–85 min | 0.78 | 0.73 | 0.29 | 0.60 | 0.91 | 0.48 |
| 85–90 min | 0.78 | 0.73 | 0.29 | 0.60 | 0.91 | 0.48 |

Figures 8 and 9 show the heat map of truck speed and truck flow in each road section of the three algorithms. As can be seen in Figure 8, in the second half of the road, that is, after link 10, the overall speed of the trucks in the DFCM-RDNSGA-III algorithm is slightly better than the other two algorithms. The main reason is that the DFCM-RDNSGA-III algorithm performs relatively better in the control of the last two ramps, which ensures the speed of the trucks on the main road. As can be seen in Figure 9, the performance of the heat map of the truck flow on the aspect of the three algorithms is basically the same as that of the car flow. The flow of the NSGA-III algorithm changes greatly before and after link 10. Compared with the RDNSGA-III algorithm, the DFCM-RDNSGA-III algorithm is smoother in the flow transmission and has a gradual increase process. In general, the control effect of the DFCM-RDNSGA-III algorithm is better than the other two algorithms.

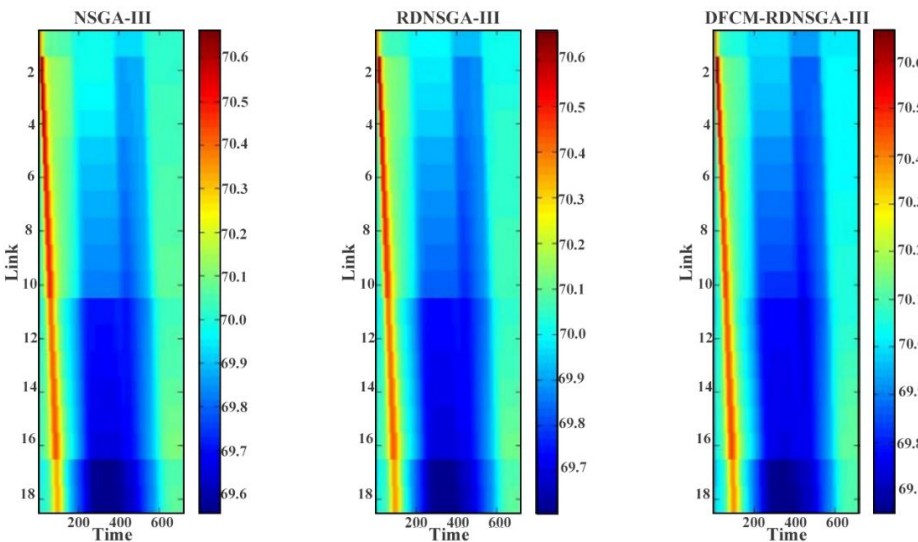

**Figure 6.** Car speed in each road section of three algorithms.

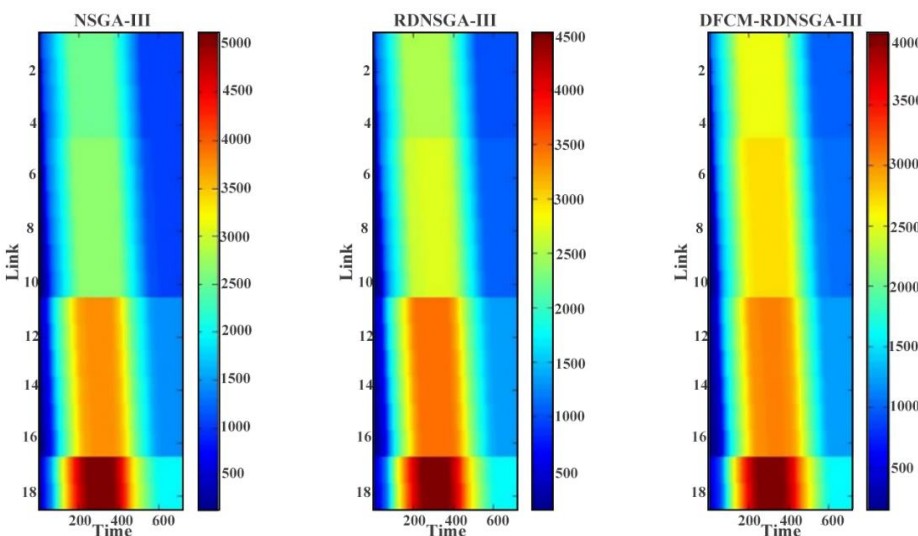

**Figure 7.** Car flow in each road section of three algorithms.

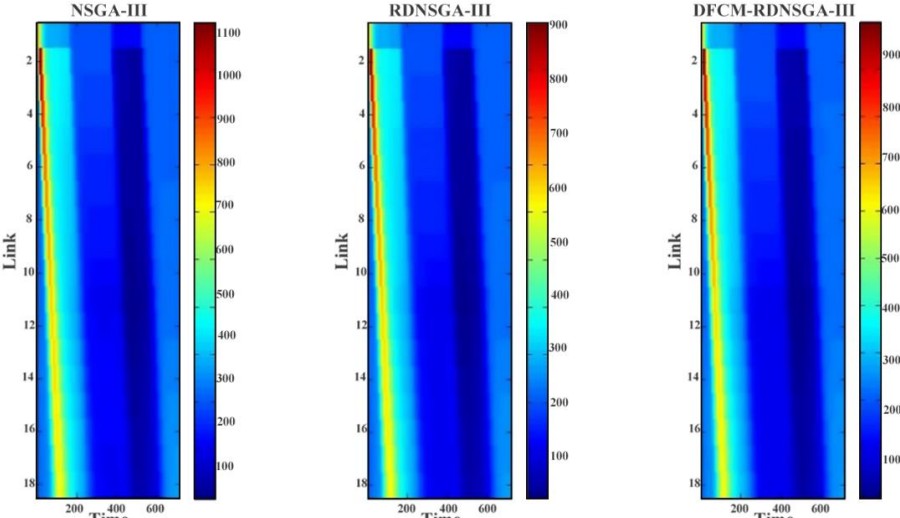

**Figure 8.** Truck speed in each road section of three algorithms.

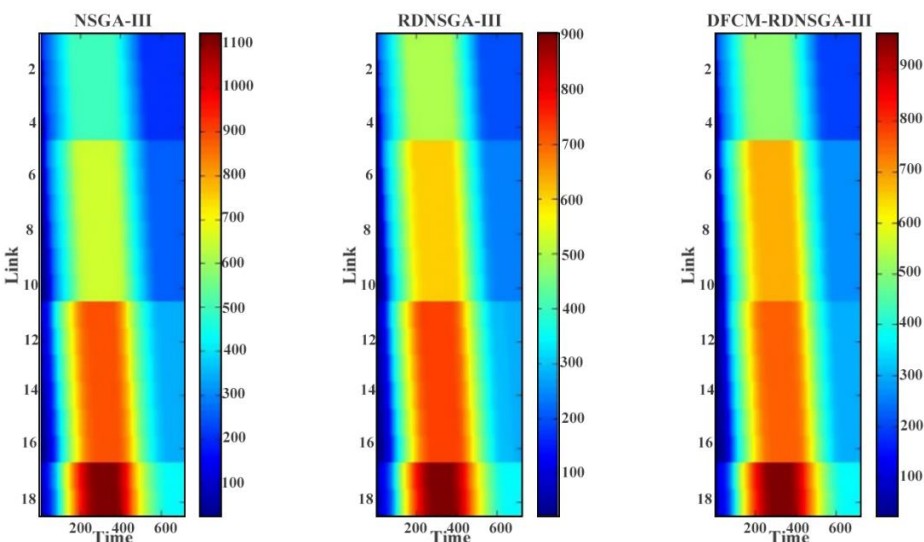

**Figure 9.** Truck flow in each road section of three algorithms.

## 6. Conclusions

In this paper, the robust dynamic traffic multi-objective optimization problem of a multi-class expressway is studied. By considering three traffic indicators—travel time, ramp queue and traffic emissions—at the same time, the implementation of variable speed limits and ramp metering strategies on an urban expressway is explored in this paper. Based on the multi-class macroscopic traffic flow model multi-class METANET and the multi-class emission and fuel consumption model multi-class VT-macro, a multi-on-ramp and multi-off-ramp expressway model is proposed. The multi-on-ramp and multi-off-ramp expressway system optimization problem is described as a robust dynamic multi-objective optimization problem. The DFCM-RDNSGA-III algorithm is proposed to solve the variable speed limit and ramp metering scheme under this problem, and the algorithm effect is verified based on VISSIM simulation. In general, compared with NSGA-III and RDNSGA-III, DFCM-RDNSGA-III, a robust dynamic genetic algorithm based on density fuzzy c-means clustering method, can better deal with the robust dynamic optimization problem which considers environmental factors and traffic congestion.

In future research, the robust dynamic multi-objective optimization algorithm needs to be improved in the following aspects: the high evolution pressure when the dimension is high, the insufficient factors considered in the evolution process to adapt to the complexity of high-dimensional problems, the irregular Pareto front exploration, and the balance between dynamic capabilities and robustness. Moreover, the existing traffic flow model does not match the actual traffic operation completely. Factors affecting the traffic condition must be studied from more aspects, and a traffic flow model better matching the actual road network must be designed.

**Author Contributions:** Funding acquisition, J.C.; methodology, J.C.; project administration, J.C.; software, Q.F. and Q.G.; supervision, J.C.; validation, Q.G.; visualization, Q.F. and Q.G.; writing— original draft, J.C.; writing—review & editing, Q.F. and Q.G. All authors have read and agreed to the published version of the manuscript.

**Funding:** This research was funded by National Natural Science Foundation of China, grant number 61104166.

**Conflicts of Interest:** We declare there are no conflict of interest regarding the publication of this paper. We have no financial and personal relationships with other people or organizations that could inappropriately influence our work.

## Appendix A

**Table A1.** Acronym List.

| Acronym List | |
| --- | --- |
| **English Abbreviation** | **English Full Name** |
| VSL | variable speed limit |
| RM | ramp metering |
| DOPs | dynamic optimization problems |
| DFCM | density fuzzy c-means clustering algorithm |
| NSGA-III | fast and elitist multiobjective nondominated sorting genetic algorithm |
| RDNSGA-III | robust dynamic nondominated sorting multi-objective genetic algorithm |
| DFCM-RDNSGA-III | robust dynamic nondominated sorting multi-objective genetic algorithm based on density fuzzy c-means method |
| SC | silhouette coefficient |
| DBI | Davies–Bouldin index |

**Table A2.** Notation List.

| Notation List | |
| --- | --- |
| **Notation** | **Meaning** |
| $T$ | length of temporal windows |
| $A$ | number of the temporal windows |
| $c$ | vehicle classes |
| $i$ | index for freeway section |
| $o$ | index for freeway on-ramps |
| $k$ | simulation time step counter |
| $l_{o,i,c}$ | queue length of class $c$ in section $i$ of on-ramp o |
| $\rho_{i,c}$ | traffic density of class $c$ in section $i$ |
| $L_i$ | length of section $i$ |
| $\lambda_i$ | number of lanes in section $i$ |
| $y$ | set of emission categories |
| $J^t_{y,i,c}$ | emissions generated by class $c$ travelling in section $i$ at normal travelling state |
| $J^s_{y,i,c}$ | emissions generated by class $c$ waiting in section $i$ at stopping state |
| $J_{y,on,o,c}$ | emissions generated by class $c$ at on-ramp $o$ |
| $l_{i,c}$ | queue length of vehicle class $c$ at on-ramp $i$ |
| $d_{i,c}$ | allowed traffic volume of class $c$ entering mainstream at on-ramp $i$ |
| $r_{i,c}$ | actual traffic volume of class $c$ entering mainstream at on-ramp $i$ |
| $\bar{r}_{i,c}$ | demanded on-ramp traffic volume of class $c$ entering section $i$ |
| $\mu_{i,c}$ | portion of the flow allowed to enter the mainstream under ramp |
| $\sigma$ | threshold of detection |
| $q_{i,c}$ | traffic volume of class c entering section $i$ |
| $VSL_{c,i}$ | variable speed limit of vehicle class $c$ in section $i$ |
| $t$ | environmental detection counter |

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
