# Peer review of "Multi-Class Freeway Congestion and Emission Based on Robust Dynamic Multi-Objective Optimization"

_algorithms, doi:10.3390/a14090266_

Round 1
Reviewer 1 Report
Some issues need to be considered:
- Please provide the list of abbreviation and symbol.
- What is the reason of using FCM, instead of other clustering algorithm?
- I think it would be better if the authors provide a block diagram or/and flowchart to represent the proposed method.
- How to determine the parameter settings in Tables 1, 2, 3? If the parameters are changed, how about the performances? Is it possible if there is an interval that can represent the best value of each parameter?
- How many cluster was used in the paper?
- How many times was the experiment run? Were the result of the travel time, ramp queue, and traffic emission, the worst, average, or the best result?
- How about the computation time and also the comparison with other methods?
Author Response
Dear professor:
Thanks very much for reviewing the paper and proposing interesting suggestions.
We have revised the paper and answered the points according to the comments carefully. Please take a look at it.
Sincerely yours
Chen Juan
Reviewer 2 Report
A robust dynamic nondominated sorting multi-objective genetic algorithm II based on density fuzzy c-means clustering to solve the problem of traffic congestion and emission optimization of urban multi-class expressway is proposed. The goals of this research and its contribution to the state of the art in the literature are clearly and comprehensively highlighted.
However, an analysis of the computational complexity of the algorithm is lacking; it is essential in order to fully evaluate its performance.
In section 4 it is necessary to present the proposed algorithm in structured mode. Authors must show the RDNSGA-III and DFCM-RDNSGA-III algorithms in pseudocode mode, rather than in consecutive steps. Furthermore, a figure is also needed that schematises the framework of the DFCM-RDNSGA-III algorithm in its functional components as a flow diagram.
rom the simulations it emerges that the preformances of RDNSGA-III and DFCM-RDNSGA are comparable. This result shows that it is necessary to highlight the computational complexities of the two algorithms and their execution times and memory consumption to detail the added performance value brought by DFCM-RDNSGA and what may be the weaknesses. Authors must complete this comparative analysis and specify in depth what the pros and cons of DFCM-RDNSGA are.
Author Response

(The authors gave the same response as above.)

Round 2
Reviewer 1 Report
4. How to determine the parameter settings in Tables 1, 2, 3? If the
parameters are changed, how about the performances? Is it possible
if there is an interval that can represent the best value of each
parameter?
5. How many clusters were used in the paper?
I can't find the answers to the 4th and 5th questions. For the 4th question, the answer should be close to Tables 1, 2, and 3, but I didn't find any highlights there.
For the other questions, the authors had answered well.
Reviewer 2 Report
The authors have improved the quality of this manuscript, taking into account all my suggestions. I consider this paper publishable in the current form.